# Modulation of LPS-Induced Neurodegeneration by Intestinal Helminth Infection in Ageing Mice

**DOI:** 10.3390/ijms241813994

**Published:** 2023-09-12

**Authors:** Natalia Jermakow, Weronika Skarżyńska, Katarzyna Lewandowska, Ewelina Kiernozek, Katarzyna Goździk, Anna Mietelska-Porowska, Nadzieja Drela, Urszula Wojda, Maria Doligalska

**Affiliations:** 1Faculty of Biology, University of Warsaw, Miecznikowa 1, 02-096 Warszawa, Poland; n.jermakow@student.uw.edu.pl (N.J.); w.skarzynska@cent.uw.edu.pl (W.S.); e.kiernozek@uw.edu.pl (E.K.); kj.gozdzik@uw.edu.pl (K.G.); n.drela@uw.edu.pl (N.D.); 2Faculty of Chemistry, Nicolaus Copernicus in Toruń, Gagarina 7, 87-100 Toruń, Poland; reol@chem.umk.pl; 3Laboratory of Preclinical Testing of Higher Standard, Nencki Institute of Experimental Biology, Ludwika Pasteura 3, 02-093 Warszawa, Poland; a.mietelska@nencki.edu.pl (A.M.-P.); u.wojda@nencki.edu.pl (U.W.)

**Keywords:** helminths, persistent infection, neuroinflammation, ageing, neurodegenerative diseases, immunomodulation, neurobehavioural manifestations

## Abstract

Parasitic helminths induce a transient, short-term inflammation at the beginning of infection, but in persistent infection may suppress the systemic immune response by enhancing the activity of regulatory M2 macrophages. The aim of the study was to determine how nematode infection affects age-related neuroinflammation, especially macrophages in the nervous tissue. Here, intraperitoneal LPS-induced systemic inflammation resulting in brain neurodegeneration was enhanced by prolonged *Heligmosomoides polygyrus* infection in C57BL/6 mice. The changes in the brain coincided with the increase in M1 macrophages, reduced survivin level, enhanced APP and GFAP expression, chitin-like chains deposition in the brain and deterioration behaviour manifestations. These changes were also observed in transgenic C57BL/6 mice predisposed to develop neurodegeneration typical for Alzheimer’s disease in response to pathogenic stimuli. Interestingly, in mice infected with the nematode only, the greater M2 macrophage population resulted in better results in the forced swim test. Given the growing burden of neurodegenerative diseases, understanding such interactive associations can have significant implications for ageing health strategies and disease monitoring.

## 1. Introduction

Neurodegenerative diseases significantly impair the quality of life in older individuals. The imbalance between anti-inflammatory and pro-inflammatory responses in neural tissue accompanies Alzheimer’s disease (AD), which results in atrophy of the cerebral cortex. Persistent neuroinflammation induced by genetic and environmental factors may provide irreversible loss of neural structure and function. Progressive neurodegeneration, cognitive deficits and behavioural manifestations such as anxiety and depression are symptoms of the disease. Anti-inflammatory treatment is associated with reduced AD prevalence [1], and pharmacological interventions may prevent anxiety-related disorders [2]. The complexity of molecular mechanisms accompanying brain pathogenesis requires in-depth research on model animals and may provide a better understanding of the role of immune processes in the ageing brain.

Neurodegenerative diseases are associated with systemic and local intra- and extracellular changes; activated astrocytes and microglia produce nitric oxide (NO), reactive oxygen species and pro-inflammatory cytokines. IL-1β, IL-6 and TNF-α levelsperipherally in the body and in the brain [3,4]. Glial-cell-derived mediators increase the level of cytokines, while immune cells of peripheral origin have been identified during neuroinflammation [5,6]. Oxidative stress contributes to neurodegeneration by causing damage to axons and neurons. In addition to oxidative stress and immune-mediated inflammatory responses, glutamate excitotoxicity and mitochondrial dysfunction also play a role in the pathogenesis and progression of neurodegenerative diseases. During demyelination, the concentrations of 5-hydroxytryptamine, tryptophan (TRP) and kynurenine (KYN) metabolites are changed [7] and contribute to the development of pathological conditions, including neurological and psychiatric disorders [8]. The increased levels of the KYN metabolite in the brain have been associated with altered fear states resulting from trauma, stress and anxiety [9].

In AD, the aggregation of misfolded proteins, β-amyloid (Aβ) and tau initiates a local inflammatory cell response in the central nervous system (CNS). There are extracellular deposits of amyloid plaques, mostly composed of Aβ, derived from the proteolytic cleavage of amyloid precursor protein (APP) [10]. Macrophages can effectively reduce Aβ deposition in the brain [11]. Fast clearance of Aβ by phagocytic cells may inhibit the release of pro-inflammatory factors that support neuronal survival and axonal regeneration. On the other hand, as Aβ aggregates, it leads to signalling impairments causing the cells to undergo apoptosis. The increasing number of apoptotic cells, inflammation and phagocytosis characterise the exacerbated pathology in brain disease [12].

Astrocytes may undergo astrogliosis. The growing amount of glial fibrillary acidic protein (GFAP) is accompanied by progressive changes in the cytoskeleton of these cells. GFAP directly affects age-related astrocyte dysfunction in neurodegeneration and is a driver of neuronal and synaptic dysfunction [13].

In recent years, immune processes regulated by helminth infections have received increasing attention for anti-inflammatory therapy. Some parasite-derived factors are potential candidates for the treatment of chronic diseases with different immune backgrounds [14,15]. In laboratory models, reduction and even remission of several diseases including experimental autoimmune encephalomyelitis (EAE, a model of multiple sclerosis) have been demonstrated [16,17,18,19]. The impact of long-lasting nematode infection on neuroinflammatory processes has not been studied.

In the experimental animal model of AD, LPS injections stimulate microglia and cause systemic inflammation and neuroinflammation with an increase in Aβ level and neuronal cell death. In the mouse brain, Aβ42 clearance is attributable primarily to microglia [20]. Both microglia and neurons form N-acetyl-D-glucosamine polymers (GlcNAc), which cause neurotoxicity in vitro [21]. In AD patients, chitin-like glucosamine polymers were elevated and accumulated in the brain, providing a scaffold for Aβ deposition in brain atherosclerotic plaques [22,23,24]. Chitin synthases are active in nematodes during eggshell production, which may have a cumulative impact on host physiology, especially in chronic nematode infections [25,26]. Whether the prolonged infection results in chitin-like polymer deposition in mice’s brain parenchyma is unclear.

Shifting the activity of brain-resident macrophages from M1 to M2 may modulate neurodegenerative diseases [27]. In AD, blood-derived monocyte/macrophages invade the CNS [28], and both M1 and M2 macrophages are associated with the progression of the disease [29]. M1 macrophages releasing pro-inflammatory mediators are associated with tissue damage. M2 macrophages releasing anti-inflammatory mediators are associated with tissue recovery [30].

*Heligmosomoides polygyrus* infection induces strong type 2 immune responses and M2 macrophage activation, minimizing excessive inflammation by regulatory mechanisms [31]. In aged *H. polygyrus-*infected mice, the shift to Th2 immune response manifested by IL-4 level was significantly lower than in young individuals [32]. Survivin facilitates the development of biased Th2 polarization by promoting the expression of interleukin 4 (IL-4) [33]. In the ageing brain, astrocytes regulate the progenitor cell cycle by affecting survivin production, which restores cell proliferation [34,35]. Survivin expression is higher in LPS (classically)-activated macrophages than in IL-4 (alternatively)-activated macrophages and therefore may mark the intensity of inflammation in the brain [36]. The relationship between the level of survivin in the brain and LPS-induced inflammation in the context of neural-immune regulation and stress-related behaviour during prolonged nematode infection is unclear. *H. polygyrus* infection may skew the phenotype of macrophages from a pro-inflammatory to an anti-inflammatory state. Extracellular vesicles secreted by the nematode and internalised by macrophages cause downregulation of type 1 and type 2 immune response-associated molecules [37]. In the chronic phase of *H. polygyrus* infection, hyporesponsiveness of lymphocytes and inhibited apoptosis correlated with the overexpression of survivin and antiapoptotic protein Bcl-2 in CD4+ T cells [38]. The relationship between survivin conditions in pathology and the exact role of macrophages in the reduction of inflammation in the nervous tissue in mice infected with the intestinal nematode has not been explored nor reported.

This study aimed to determine whether chronic *H. polygyrus* infection in LPS-inflamed mice would ameliorate the development of neurodegenerative changes in the brain. We measured the impact of long-term *H. polygyrus* infection on pathological lesions in LPS-induced neurodegeneration in C57BL/6 strain mice and B6; SJL-Tg(APPSWE)2576Kha transgenic mice, which represent AD pathology in response to pro-inflammatory stimuli. Primary LPS-induced inflammation was prolonged by the nematode infection and worsened selected manifestations of neurodegeneration.

We used a laboratory model to explore the macrophage-related mechanisms for modulation of LPS-induced neuroinflammation during chronic infection with the intestinal nematode. Our aim was to highlight the complex and overlapping mechanisms involved in the development of neurodegeneration and the emergence of depressed behaviour in mice. We encountered several limitations of the proposed model while suggesting a new direction for research in transgenic mice. We studied nematode infection to assess the modulation of neurogenerative pathology. Further research is needed to elucidate the mechanisms responsible for maintaining inflammation in the brain during chronic nematode infection.

## 2. Results

### 2.1. Inflammatory Leucocyte Count in the Blood

Neutrophils, monocytes and eosinophils were monitored for 95 days post-infection (DPI), and the percentage of cells was evaluated in venous blood smears (Figure 1). Before treatment, the percentage of neutrophils oscillated between 12 and 15%. Infection with *H. polygyrus* increased the level by more than 45% on the 5th day PI in both the HP and LPS + HP groups (the difference between the percentage of neutrophils in groups HP vs. LPS was statistically significant at *p* < 0.006, and in groups, LPS + HP vs. LPS at *p* < 0.004). Ten days later, neutrophils reached the highest level as the percentage increased to above 50% in all examined groups. The difference between groups HP vs. LPS and LPS + HP vs. LPS was still significant at 35 DPI (*p* < 0.017 and *p* < 0.001, respectively). Endotoxin enhanced neutrophil percentage in the blood. Prolonged infection was accompanied by a neutrophil expansion in mice injected with LPS and infected with the nematode (LPS + HP) and in mice only infected with *H. polygyrus* (HP) (ANOVA: between groups, *p* < 0.002, the interaction between group and day, *p* < 0.0001). In mice infected with *H. polygyrus* (HP and LPS + HP), the percentages of cells were higher than in mice treated with LPS alone (LPS). The neutrophil, monocyte and eosinophil increase is characteristic of systemic and prolonged inflammation.

Interestingly, the percentage of monocytes in the blood gradually decreased in elderly mice (ANOVA: between day, *p* < 0.000001). The highest percentage (5.6%) of these cells was present in mice injected with LPS and infected with *H. polygyrus* on day 25 post-infection (LPS + HP). The difference between groups LPS + HP and LPS was significant at *p* < 0.04.

The percentage of eosinophils in the blood remained elevated till 25 DPI (statistically significant difference between HP and LPS was at *p* < 0.012, and between LPS + HP and LPS at *p* < 0.045).

In mice injected with LPS, the neutrophil percentage increased, the monocyte percentage gradually decreased and eosinophils around the control level. Infection with *H. polygyrus* induced a monocyte and eosinophil response in the blood till 25 DPI and maintained a high neutrophil percentage for a prolonged time in mice older than 8 months.

### 2.2. Changes in the Macrophage Population in the Cerebrospinal Fluid

Cerebrospinal fluid (CSF) surrounds the brain and spinal cord. FACS analysis was used to determine the percentage of M1 and M2 macrophages in the cerebrospinal exudate in the examined mice (Figure 2A–C).

The percentage of M1 (CD206+ CD80+) macrophages was very low, and changes in their percentage were slightly affected by the treatment. Significant differences in the M1 macrophage percentages were observed between the control (CTR) and the other groups (CTR vs. LPS, *p* < 0.005; CTR vs. HP, *p* < 0.0004; CTR vs. LPS + HP, *p* < 0.0007 and CTR vs. APPSWE + HP, *p* < 0.0017). In control mice (CTR), M1 consisted of no more than 0.5%, but in LPS-treated mice (LPS and LPS + HP groups), the percentage of M1 macrophages oscillated between 1.18 and 1.32 (Figure 2A) as in mice infected with *H. polygyrus* (HP and APPSWE + HP groups), where the percentage of M1 was 1.3 and 1.2%. Significant differences in the M1 macrophage percentages were observed between the control group (CTR) and the other groups (*p* < 0.05).

The M2 macrophages dominated in all groups. In control mice, M2 (CD206 + CD80−) macrophages comprised 20% of the total cell population. In mice treated with LPS, M2 cells increased to 30% and in mice infected with *H. polygyrus* (HP) to nearly 60%. In the groups LPS + HP and APPSWE + HP, the percentage of M2 oscillated around 45%. Significant differences in the contents of M2 macrophages were observed between the control group (CTR) and the other groups (CTR vs. LPS, *p* < 0.0072; CTR vs. HP, *p* < 0.00001; CTR vs. LPS + HP, *p* < 0.00001 and CTR vs. APPSWE + HP, *p* < 0.00001). The difference between groups HP vs. LPS was statistically significant at *p* < 0.00001 and also between groups HP vs. LPS + HP (*p* < 0.0265) or HP vs. APPSWE + HP (*p* < 0.0185). In comparison to LPS groups, infection with *H. polygyrus* raised the percentage of M2 in groups LPS + HP (*p* < 0.0005) and APPSWE + HP (*p* < 0.0013, ANOVA: between group, *p* < 0.00001).

### 2.3. Survivin Level in the Brain Homogenate

The activity of astrocytes determines the survivin concentration in the brain [35]. In mice infected with *H. polygyrus* (groups HP and LPS + HP), survivin concentration decreased (Figure 3). The highest level of survivin was present in control C57BL/6 mice (250 pg/mL). The lowest level of this protein was observed in mice treated with LPS and infected with *H. polygyrus* (92.8 pg/mL in group LPS + HP); the concentration dropped more than two times in comparison to the control mice (CTR vs. LPS + HP, *p* < 0.0001). Interestingly, *H. polygyrus* infection inhibited survivin production (CTR vs. HP, *p* < 0.0032, LPS vs. HP, *p* < 0.0047, LPS vs. LPS + HP, *p* < 0.0001). The concentration of survivin in the group LPS + HP was lower than in the group HP (*p* < 0.0016) and APPSWE + HP (*p* < 0.0002, ANOVA: between group, *p* < 0.00001). The survivin production was not affected by *H. polygyrus* infection in APPSWE mice.

### 2.4. Chitin-Like Polysaccharides and FTIR Analysis

Selective staining of tissue was used to analyse the presence of chitin-like polysaccharides in the brain. Fluorescent Brightener 28 showed enhanced fluorescence when bound to the tissue. Structures of different sizes fluorescing at different intensities were detected in the extracellular space (Figure 4). The bright artefacts were present in mice infected with the nematode, particularly in the LPS + HP group. In control mice, there was no labelling in tissues processed for Calcofluor staining (Figure 4A). APPSWE + HP mice were positive for chitin-like polysaccharides.

In light of the positive reaction for chitin-like polysaccharides in samples, attenuated total reflection (ATR) Fourier transform infrared (FTIR) spectroscopy was used to detect chitin in the brain homogenate (Figure 4B,C). The analysis confirmed the presence of chitin-like polysaccharides with spectra typical for chitin; comparing the spectra of chitin (Chit) and D-GlcNAc, a similar pattern of FTIR spectra was observed in all groups. FTIR analysis showed the amide I at 1648 cm^−1^ and peak. The absorption bands of amide II and III in chitin samples were identified at 1542 cm^−1^ and 1307 cm^−1^, respectively. The amide I, amide II and amide III peaks clearly showed that the obtained three-dimensional chitin is in the form of alpha crystallization.

In comparison to the chitin spectra pattern, peaks ranging between 2000 and 2300 cm^−1^ were expressed with the highest intensity in the group LPS + HP. Chitin-like polysaccharides were deposited in mice infected with *H. polygyrus* in the brain tissue.

### 2.5. Analysis of APP and GFAP Distribution in the Brain

Immunostaining of APP and GFAP was performed on the cross-section of the brain on the same slides as a double reaction and then counterstained with H&E. The proteins in the brain in C57BL/6 mice and in APPSWE mice are shown in Figure 5. The pictures at the adult 3D coronal fibre tracts level showed APP and GFAP. The double immunostaining for APP and GFAP in the right hemisphere showed only a few positive cells in the control group (CTR) in the hill area (Figure 5A) and in the hippocampus area (Figure 5B). In the LPS group, single-positive nerve cells were found in the thalamus (Figure 5C), the motor cortex (Figure 5D) and the piriform cortex (Figure 5E). In mice infected with *H. polygyrus*, weakly labelled positive neural cells were present near the cortical structures. They were positive in the sensory cortex (Figure 5F), the motor cortex (Figure 5G) and the piriform cortex (Figure 5H). Due to the selective deposition of APP (especially in neurons) and GFAP (mainly in astrocytes) in dissimilar nervous tissue cells, the results indicate the accumulation of only a few APP but not GFAP in neurons (Figure 5F–H). In turn, the most intensively stained astrocytes for GFAP were present in LPS + HP and APPSWE + HP groups (Figure 5I–N).

### 2.6. Western-Blot Analysis of APP and GFAP in the Hippocampus and Cortex

The immunohistochemical identification was confirmed by Western blot analysis (Figure 6). In mice injected with LPS at 4 months of age, small chains of APP protein were found 8 months later in the hippocampus and also in the cortex (Figure 6A,C,D). As the amount of the long chain (100 kDa) decreased, the short chain (24–28 kDa) increased in LPS + HP (Figure 6A,D). Additional bands were observed in the hippocampus of mice, infected with *H. polygyrus* (HP) and in the LPS + HP group (Figure 6A,C). In mice injected with LPS, APP subunits were visualised as the band with a relative molecular mass of 24 kDa (Figure 6A, C). In mice infected with *H. polygyrus*, two bands were present with relative molecular masses of 24 kDa and 28 kDa. The positive results were also observed in mice of the LPS + HP group. The strongest signal was associated with an additional band of 100 kDa relative molecular weight, which was also typical for mice in the APPSWE + HP group. In the brain cortex, APP appeared in all examined mice; one wide band of 100 kDa relative molecular weight with high expression intensity was seen in all treated mice (Figure 6A,C).

Analysis of GFAP distribution in the brain showed that the protein was present in the hippocampus in a small amount, while it localised mainly in the cortex. The level of the protein increased preferentially in LPS + HP and APPSWE + HP mice. The relative molecular mass of this protein was 50–52 kDa (Figure 6B) in both the hippocampus and the cortex.

In mice exposed to LPS and *H. polygyrus* infection, the amount of cleavage proteins increased both in the hippocampus for APP and in the hippocampus and cortex for GFAP. The most intense accumulation of proteins was present in mice of the LPS + HP and APPSWE + HP groups.

The pathogenic changes were confirmed in the local distribution of proteins in the brain (Figure 5). APP and GFAP-positive spots appeared in the brains of mice infected with *H. polygyrus*. In C57BL/6 mice injected with LPS and infected with the nematode (LPS + HP), the pathologic changes were intensified. Also, APPSWE + HP mice were positive for APP and GFAP.

Several bands were immunoreactive with the antibody against the glial fibrillary acidic protein (GFAP), a cytoskeleton filamentous protein specific to astroglial cells (Figure 6B,E). Protein bands were detected around the molecular mass of 50 kDa and greater (around 100 kDa) in LPS-treated and *H. polygyrus*-infected mice. Only a few bands around the same region were found in the cortex of control mice (CTR). In treated mice, these bands were preferentially present in the cortex and also in the hippocampus.

### 2.7. Behavioural Observations in the Forced Swimming Test

Figure 7 shows selected images of mice manifesting motor skill disorders measured as the average number of seconds (±standard error) that the mouse remains stationary. APPSWE + HP mice remained still for the longest time among all tested groups (Figure 7A: APPSWE + HP vs. CTR, *p* < 0.0009). Mice in the APPSWE + HP group exhibited problems with coordination of movements during attempts to stay afloat, as observed during visual assessment and they stayed immobile for 29 s. Coordination abnormalities were also found in mice in the LPS + HP group (18.8 s, LPS + HP vs. CTR, *p* < 0.005). Mice of the HP group remained immobile for the shortest time, less than 4 s. Mice of the HP group performed better in the test than the control uninfected mice aged 12 months (Figure 7B, HP vs. CTR, *p* < 0.00001) (ANOVA: between group, *p* < 0.00001).

In summary, following treatment with LPS, there was an approximately four-fold increase in the percentage of neutrophils from about 12–15% to 50% of the total (Figure 1). There was a similar increase in the neutrophil percentage in the animals infected with *H. polygyrus*; the percentage peaked at the same time. Neutrophil kinetics were similar in the mice treated with LPS and infected with nematodes to the mice only infected with nematodes. LPS only affected neutrophil percentages, while nematode infection also increased the proportion of monocytes and eosinophils. In the CSF, most macrophages were of M2 phenotype. Following LPS treatment, the percentage of M2 macrophages increased from 20 to 30% (Figure 2). There was a larger increase in the nematode-infected mice to nearly 60% while mice given both treatments had an intermediate percentage of M2 macrophages in the CSF. The concentration of survivin in the brain fell following nematode infection and was even greater in mice given LPS and nematodes (Figure 3). Chitin-like polysaccharides were present in the nematode-infected mice, and fluorescent staining was even stronger in the mice given LPS treatment and infected with nematodes (Figure 4). GFAP and APP staining was not present in the brains of control mice, relatively weak in nematode-infected mice or mice given LPS and strongest in mice given both LPS and nematode infection (Figure 5). Western blotting confirmed the presence of cleaved APP in the hippocampus and GFAP in the hippocampus and cortex of treated mice. The mice given both LPS and nematodes showed higher levels of these proteins than control mice or mice given a single treatment (Figure 6). Finally, a forced swim test was used to assess deterioration in motor skills following treatment. Mice given LPS showed slightly longer freezing times than control mice, but nematode-infected mice had substantially shorter freezing times than control mice. Animals infected with *H. polygyrus* were resistant to stress conditions produced in the forced swim test.

## 3. Discussion

Inflammation caused by many pathogens or environmental factors can severely disrupt homeostatic processes in the host. In ageing individuals, prolonged immune mobilization affects physiological and mental activities. Systemic inflammation may also project an immune reactivity in the nervous tissue with the outcome of neuropathology.

We studied whether long-term infection with intestinal nematodes can inhibit neurodegenerative processes in the brain of ageing C57BL/6 mice, induced by LPS treatment in adulthood. Signs of systemic inflammation in the blood and locally in brain tissue were examined in mice. We tested the hypothesis of inhibition of inflammation in the brain by *H. polygyrus* infection. We assumed that the ongoing inflammation may contribute to the development of Alzheimer’s disease, hence investigating whether the changes after LPS treatment have the characteristics of AD. SJL-Tg(APPSWE)2576Kha transgenic mice are predisposed to express AD symptoms, and we compared the effect of *H. polygyrus* infection on neurodegenerative changes in both strains of mice.

The systemic inflammatory response, monitored by the percentage of cells in the blood, was dominated by neutrophils. In addition to the neutrophil response, an increase in monocyte and eosinophil percentages was observed in nematode-infected mice, which coincided with the localization of larvae and adult forms in the wall or in the lumen of the intestine, respectively. Locally, inflammation starts at the early stage of granuloma formation in the intestinal wall occupied by developing nematode L4 stage larvae, which are eliminated by antibody-dependent cell-mediated cytotoxicity (ADCC), mainly by macrophages and eosinophils [39,40,41]. In the later stage of granuloma formation, alternatively activated macrophages (AAMs) and eosinophils produce immunoregulatory and wound-healing molecules [42]. Immunity to intestinal lumen adult worms is regulated by IL-4 and IL-13, controlling intestinal muscle contraction [43,44] and antibody production. Induced immunoregulatory mechanisms allow adult worms to persist in the intestine [16]. To our knowledge, the neutrophil response accompanying *H. polygyrus* infection has not been measured intensively, especially in ageing mice. We found that moderate inflammation accompanying prolonged *H. polygyrus* infection in mice is characterised mainly by neutrophil involvement. In other studies, chronic *H. polygyrus* settlement in the intestine enhanced acute airway neutrophil responses to *P. aeruginosa* infection [45]. Primary *H. polygyrus* infection induces a neutrophil-like subset of myeloid-derived suppressive cells that inhibit Th2 responses and therefore promote chronic infection [46]. These results suggest that neutrophil responses might be enhanced during prolonged nematode infection.

A progressive decrease in the percentage of circulating monocytes in the blood was observed in all examined older mice. The changes might reflect impaired inflammatory monocyte development or premature egress from blood vessels [47]. The reduced monocyte response was ineffective at *H. polygyrus* removal; nematode egg production continued until 28 weeks post-infection. The role of neutrophils and monocytes needs to be better characterised during persistent *H. polygyrus* infection in ageing mice. Changes in the blood cell percentage in mice show that the inflammatory response induced by LPS was not inhibited by persistent *H. polygyrus* infection. The neutrophil response induced by *H. polygyrus* supported LPS-dependent neutrophilia.

Examination of cerebrospinal fluid reflected the domination of M2 over M1 macrophages. M1/M2 polarization of macrophages plays an important role in controlling the balance between induction and suppression of inflammation. The percentage of M1 macrophages was reduced compared to the M2 population in all ageing mice, and the increase in LPS-injected or *H. polygyrus*-infected mice was significant. The population of M2 macrophages in CSF increased significantly in elderly mice infected with *H. polygyrus*.

In CNS, resident macrophages–microglia and CNS-associated macrophages triggered by environmental factors in the brain parenchyma undergo differentiation and maturation. Also, macrophages, as bone-marrow-derived circulating monocytes, can enter the brain and replace resident cells where they participate in pathogen elimination or tissue damage [48]. In pathological conditions, microglia and macrophages, derived from circulating monocytes, constitute the first line of defence and regulate both innate and adaptive immune responses [49]. M1-polarised microglia can produce pro-inflammatory cytokines, reactive oxygen species and nitric oxide, suggesting that these molecules contribute to the dysfunction of neural networks in the CNS. Alternatively, M2-polarised microglia express cytokines and receptors that promote the inhibition of inflammation and restore homeostasis [50]. LPS, acting as a pro-inflammatory factor, and *H. polygyrus*, as a source of both pro-inflammatory and anti-inflammatory factors, might activate different macrophage populations and influence their functions. The M2 macrophage population response is typical for *H. polygyrus* infection and depends on IL-4/IL-13, cytokine production [39,51,52]. Significant mobilization of M2 macrophages in LPS-treated and infected mice may be related to the induction of immune regulation by the parasite. They not only counteract systemic inflammation but also support adults to survive in the intestine and maintain healing reactions [53,54,55]. However, the brain tissue is sensitive to IL-4. Even short-term IL-4 expression in the hippocampus leads to the exacerbation of amyloid deposition in vivo, possibly as a result of reduced glial phagocytosis [56,57]. The effects of the inflammatory process were shown by the highest level of GFAP in the brain homogenate and by the numerous APP/GFAP-positive cells in the brain of mice that were injected with LPS and then infected with *H. polygyrus*. In mice treated with LPS alone, the percentage of M2 macrophages increased only moderately. Several stimuli and signal pathways have been recognised as inducers of M2 activation: M2a population responds to IL-4 and IL-13, M2b is activated by immune complexes and bacterial lipopolysaccharide (LPS) and M2c responds to glucocorticoids and TGF–β [58]. In mice exposed to LPS and *H. polygyrus* infection, several sets of macrophage populations may regulate pathological changes in the brain tissue. IL-4 produced during *H. polygyrus* infection may enhance the level of LPS-dependent proinflammatory cytokines and then a greater amount of neurodegenerative products [59,60]. The most extensive and pronounced changes in the brain appeared in mice that were given LPS and then infected with nematodes. This observation prompts future studies on the role of IL-4 in nervous tissue during nematode infection.

In elderly mice with prolonged *H. polygyrus* infection, chitin-like deposits were identified in the brain. The potentially pathogenic chitin-like structure, previously described in the brains of individuals with multiple sclerosis and Alzheimer’s disease, correlated with increasing chitinase activity in inactivated microglia–macrophage cells [61]. Also, age-related secretion of chitotriosidase (chitinase 1) correlates with macrophages and neutrophil activation [62,63]. Chitin is composed of GlcNAc subunits. In mammals -GlcNAc is a basic component of hyaluronic acid (HA), which is the major component of the extracellular matrix and is extensively distributed in connective, epithelial and nervous tissues [64]. The deposition of GlcNAc chains suggests that macrophages or neutrophils are involved in neurodegeneration. In response to LPS, macrophages accumulate glutamine in vitro [65,66], and glutamine metabolism modulates the polarization of mouse M2 macrophages through the induction of the glutamine–UDP-N-acetylglucosamine pathway [67]. Glutamine metabolism to glucosamine is necessary for glutamine inhibition of endothelial nitric oxide synthesis by reducing cellular availability of NADPH [68]. The pattern of FTIR spectra in the brain suggests synthesis and/or deposition of polymer chains with GlcNAc, identified as a chitin-like crystallised structure in the tissue, especially in mice with greater amounts of APP and GFAP. The pathway for synthesis of UDP-GlcNAc is critical for M2 macrophage polarization because it is responsible for the glycosylation of macrophage mannose receptor and macrophage galactose-binding lectin, an M2 marker protein [69].

The amino sugar, GlcNAc, is synthesised de novo in the form of GlcNAc-6-*p* by eukaryotic cells. Also, the extracellular matrix of animal cells contains different sugar polymers, many of which contain GlcNAc [70]. However, the free form of GlcNAc may originate from bacterial cell wall peptidoglycan or fungal cell wall chitins [71]. During persistent infection, the changes in the brain proteome may reflect control by the parasite. In response to the accumulation of proteins, e.g., APP and GFAP, the presence of -GlcNAc chains in the brain seems to be the result of intense regulatory reactions by M2 macrophages. It would be interesting to study the influence of glutamine pathway-dependent metabolism on the polarization of macrophages from the central nervous system (microglial cells) during *H. polygyrus* infection and reflect the interactions among the host immune system, gut microbiota and the parasite. Metabolites are an important part of gut–brain communication, an axis that has been previously implicated in Alzheimer’s disease [72].

Survivin, a protein with anti-apoptotic activity, supports cell survival by controlling cell proliferation [34]. Concurrently with the increase in M2 cells, the level of survivin significantly fell upon *H. polygyrus* infection and the lowest concentration was in mice treated with LPS and infected with the nematode. Mice of that group harbour several fragments of different lengths cleaved from amyloid-β precursor protein (APP) and especially glial fibrillary acidic protein (GFAP), than other mice. GFAP is one of the best markers for the activated astrocytes following injury or stress in the CNS [73,74]. Survivin concentration depends on the activity of these cells, which in pathological conditions co-express GFAP [75,76]. Astrocyte dysfunction results in neurodegeneration [77]. The effects of the inflammatory process were shown by the highest level of GFAP protein in the brain homogenate and by the numerous APP/GFAP-positive cells in the brain of mice that were injected with LPS and then infected with *H. polygyrus*. In that group, the concentration of survivin was at the lowest level. Moreover, in mice treated with LPS and infected with *H. polygyrus*, the decrease in survivin concentration coincided with intensive deposition of APP in the hippocampus and GFAP in the hippocampus and cortex. Low levels of survivin may indicate astrocyte dysfunction or a deepening degree of pathology in the brain with the extension of the life of infected mice [35].

In our experimental model, the degree of CNS damage was assessed by the identification of proteins indicative of neurodegenerative processes in the hippocampus and cortex. The amount of APP and GFAP as a marker of neural tissue damage correlated with behavioural dysfunction, but only in mice treated with LPS. The intensive response in the brain of mice treated with LPS and infected with *H. polygyrus*, determined both by Western blot analysis and the staining of APP and GFAP-positive cells, confirmed the neurodegenerative changes. The distribution of proteins differed between the control and other groups of mice. In all elderly mice, APP was identified in the cortex; however, in the hippocampus, the protein was expressed in distinct groups. GFAP is the main astrocytic intermediate filament, and different isoforms of the protein are present in astrocytes and other niches [78]. The APPs are normal constituents of large numbers of neurons. APP metabolism has a direct effect on amyloid-β generation as Aβ is a regulated cleavage product of amyloid precursor protein [79]. Intraneuronal Aβ accumulation may be an early event in Alzheimer’s disease (AD) pathogenesis. These deposits increased with age. The earliest Aβ deposition occurs intraneuronally, prior to extracellular amyloid plaque formation [80,81]. The deposition of amyloid in the parenchyma of the brain is associated with a robust inflammatory response [82,83]. Astrocytes are tightly linked to Aβ degradation and clearance but may serve different functions as the AD progresses; astrocytes are believed to degrade and clear or even produce Aβ deposition and provide indirect neurotoxicity [84]. APP overexpression in the LPS + HP group might induce behavioural abnormalities prior to Aβ pathology [85].

The glial fibrillary acidic protein (GFAP) differentiates glial cells, so increased astrogliosis may manifest in more intense staining of astrocytes than nerve cells in LPS + HP and APPSWE + HP mice [86]. However, neurodegeneration results in both neuron (APP) and astrocyte (GFAP) staining, which represent ongoing pathological processes in the brain tissue. The results are consistent with the behaviour of mice. In the HP group, mice with very few markers of inflammation and neurodegeneration in the nervous tissue are the most resistant to stress changes in the forced swim test.

In mice genetically predisposed to brain inflammation (with Swedish K670N and M671L mutation (APPSWE, Tg4576 model), amyloid plaque deposition is associated with loss of memory and cognitive function, which is characteristic of Alzheimer’s disease [87]. Similar changes observed in the current study may have consisted of neuron dystrophy, activation of microglia cells, reactivation of astrocytes and degeneration of synaptic connections, appearing gradually with age [88]. APP protein in the hippocampus and in the cortex of APPSWE + HP mice was of the same molecular weight. This was in contrast to mice treated with LPS or infected with nematodes; these mice had several small protein subunits.

Mice treated with LPS and infected with *H. polygyrus* and mice genetically predisposed to neurodegeneration (APPSWE + HP) stored greater amounts of GFAP than other groups. Microglia and macrophages residing in the CNS might produce higher concentrations of cytotoxic factors. This proinflammatory response may facilitate the recruitment of blood-derived neutrophils and monocytes.

The consequence of LPS-induced neurodegeneration in combination with long-lasting *H. polygyrus* infection was an impaired response of mice to stress conditions. In rodent models of AD, astrogliosis is characterised by reactive astrocytes with upregulation of GFAP [88,89]. Disruption of synaptogenesis related to the pathogenetic phenotype of astrocytes strongly correlated with cognitive decline [90].

*H. polygyrus* infection induces polarization of M2 macrophages with upregulation of Arg1 and IL-10, the anti-inflammatory factors to resist metabolic inflammation mediated by M1 macrophages in mice [91]. During a single *H. polygyrus* infection, infiltrating macrophages from the circulation may facilitate the M2-like tissue remodelling response and may contribute to repair mechanisms, including re-myelination [92,93]. In mice infected with *H. polygyrus* alone, nematode molecules might skew the M2 polarization of microglia. Such an immune environment is also typical for the elderly [94]. M2 polarization may support regeneration in the brain and assist remission of psychiatric disorders [95]. Induction of regulatory mechanisms [19] resulted in a better neurological condition of mice in comparison to LPS-treated and then *H. polygyrus-*infected animals. Some neurodegenerative changes were observed mainly in mice treated with LPS and infected with nematodes; however, these were not the same as in mice used for the positive control of AD.

In conclusion, nematode infection appears to exacerbate LPS-induced encephalitis. That is in contrast to many immune disorders ameliorated by nematode infection. Further studies are needed to identify the neuroregulatory pathways induced by infection with intestinal nematodes. However, these studies suggest that treatment of immune disorders caused by nematode infection must be completed with caution, especially in elderly patients, because their immune response is slower and may cause pathogenesis in the nervous tissue [96]. Given the increasing burden of neurodegenerative diseases, understanding such connections could have important implications for health strategies to improve health and prevent dementia in ageing human populations.

## 4. Materials and Methods

### 4.1. Mice

The experiments were conducted on pathogen-free male C57BL/6 mice. Mice of strain C57BL/6 have a long (2 years) life expectancy and are used to study ageing. This strain of mice is susceptible to *H. polygyrus* infection [97]. Additionally, males of C57BL/6 mice are predisposed to mounting a more severe Th1-type immune response, while females are more likely to produce an exaggerated Th2-type response [98]. These mice exhibit a high degree of uniformity in their inherited characteristics, including behaviour, and response to experimental treatments.

As a positive control for brain AD pathology and behaviour, we used F1 C57BL/6 and B6; SJL-Tg(APPSWE)2576Kha mice. Mice were kindly given by professor Urszula Wojda from the Laboratory of Preclinical Testing of Higher Standard at the Nencki Institute of Experimental Biology, Polish Academy of Sciences and were the offspring of a heterozygous programmed genetic model of Alzheimer’s disease with a β-amyloid precursor (APPSWE mice carry a transgene coding for the 695-amino acid isoform of human Alzheimer β-amyloid (Aβ) precursor protein carrying the Swedish mutation). Mice were infected with *H. polygyrus* using the same protocol as for the other mice. In contrast to C57BL/6 mice, four APPSWE mice were kept in separate cages due to their aggressive behaviour.

The mice aged 4 months, weighed 20–25 g at the start of the study, were kept in standard light:dark conditions (12 h:12 h) at a temperature of 22 ± 2 °C with ad libitum access to water and commercial pellet food. All mice were allowed to adjust to laboratory conditions for a minimum of 7 days before experimental manipulation.

### 4.2. Schedule of the Experiment

All experiments were conducted according to the Polish Law on Animal Experimentation and EU Directive 2010/63/UE and approved by the First Warsaw Local Ethics Committee (ID 484/2017).

The experiment included the following groups: an uninfected control group (CTR), mice injected with (LPS), mice injected with LPS and infected with *H. polygyrus* (LPS + HP) and mice infected with *H. polygyrus* (HP) only. Each experimental group consisted of five animals kept together in one cage. LPS solution (*Escherichia coli* O55:B5, Merck–L2880, Jerusalem, Israel) prepared in 0.9% NaCl was administered by intraperitoneal injection for 3 days prior to nematode infection, at a total dose of 0.6 mg per mouse [3].

The mice were orally infected with 200 infective larvae (L3) of *H. polygyrus*. At the beginning of the experiment, the animals were 4 months old and were examined at 12 months. At that time, the infection was over and the parasites had been spontaneously expelled. The course of infection was monitored and the number of eggs per gram of faecal material (EPG) was counted. Starting on day 14 after infection, every week coproscopy of each mouse was performed until eggs were found to be absent in three consecutive samples, e.g., 28 weeks after *H. polygyrus* infection. Animals were sacrificed at 12 months of age and as the parasitic control, the absence of nematodes in the intestine confirmed the end of the infection.

### 4.3. Evaluation of Leukocyte Percentage in the Blood

Peripheral inflammation was monitored in the blood based on the percentage of leucocytes in the blood smear. Two hundred cells were differentiated into separate cell populations: lymphocytes, neutrophils, monocytes, eosinophils and basophils. Blood was collected from the tail every 10 days. The smears were fixed in methanol, stained with DiffQuick kit (Merck, Darmstadt, Germany) and then viewed under imaging light microscopy at 100× magnification to determine the percentage of leukocyte populations. The percentage of neutrophils, monocytes and eosinophils is presented.

### 4.4. Cell Harvesting from the Brain

The mice were anaesthetised and killed. A brain with an extended core was isolated. The brain was divided into hemispheres, the right hemisphere was preserved for histopathological examination, while the hippocampus and cerebral cortex were isolated from the left.

The brain was cooled on ice in 1 mL of phosphate buffer saline (PBS), pH 7.2 and vortexed for 30 s. The obtained cerebrospinal exudate fluid (CSF) was filtered through a BIOLGIX strainer, with a pore size of 40 μm, and centrifuged at 1200 rpm for 7 min at 4 °C. Cells were resuspended in 1% of bovine serum albumin (BSA, SigmaAldrich, St. Louis, MO, USA) in PBS, counted using an Invitrogen CountessTM automated cell counter (NanoEnTek Inc., Seoul, Republic of Korea) and used for FACs analysis.

### 4.5. Flow Cytometry Analysis of M1 and M2 Macrophages

M1 and M2 macrophages were estimated in CSF. To identify M1 and M2 macrophages, cells were stained with specific monoclonal antibodies conjugated with fluorochromes: anti-CD3/FITC, anti-CD206/APC, anti-CD11b/PerCP and anti-CD80/PE (BD Pharmingen^TM^, San Diego, CA, USA). Stained cells were examined using the FACSVerse^TM^ instrument and analysed using FACSDiva software, BD FACSDiva v9.0. The gating strategy is presented in Figure 2.

### 4.6. Identification of APP and GFAP-Positive Cells in the Brain Section

The right hemispheres of mouse brains were immersed in 10% glucose solution at −4 °C for 48 h. The brains were then placed in a dish with 2-methylbutanene, frozen in liquid nitrogen and kept frozen at −80 °C until use. Frozen brains were sliced in a microtome at −25 °C into 10 µm-thick sections and placed on L-lysine-coated glasses. Slides were fixed in ethanol and frozen at −20 °C after refreezing and were washed in Tris buffer before digestion with trypsin (0.1% Trypsin, 0.1% CaCl_2_ (Biowest, Nuaillé, France, EU) in Tris buffer, heated to 37 °C for 30 min). After washing in Tris buffer, 1% bovine serum albumin (BSA) was applied for 30 min. After excluding the presence of endogenous peroxidases, slides were prepared for immunohistochemical assays.

Immunohistochemical staining with rabbit antibodies was performed to demonstrate the cellular distribution of APP and GFAP in a cross-section of the brain. After rinsing, the sections were incubated in TRIS buffer with 2% BSA (Sigma-Aldrich, St. Louis, MO, USA) and 1% Triton X-100 (Sigma-Aldrich, USA) blocking solution for 30 min. Rabbit anti-Glial Fibrillary Acidic Protein, produced by using as immunogen a sequence corresponding to the C-terminus of human GFAP (Gene 2670), (G4546, Sigma-Aldrich, St. Louis, MO, USA) diluted 1:100 in 0.1% BSA, and rabbit anti-Amyloid Precursor Protein, Universal Polyclonal Antibody, specific to amino acids 99–126 (AB5300, MilliporeTemecula, CA, USA, Sigma-Aldrich) diluted 1:50 in 0.1% BSA, were selected for the highest dilution, yielding a positive result. Slides were washed with Tris buffer and incubated along with the anti-rabbit antibody conjugated with HRP (A-0545) (Sigma-Aldrich, Jerusalem, Israel) for 30 min and rinsed briefly with Tris buffer. The immunoreactivity was detected with the substrate, 3-3′ Diaminobenzidine tetrahydrochloride (DAB) (Vector Labs, Newark, CA, USA), then stained with 0.1% toluidine blue in Tris buffer or haematoxylin and eosin (H&E) to visualise tissue structures. The slides were embedded in 1% glycerogelatin. Preparations were viewed on a Nikon microscope at magnifications ranging from 10× to assess brain structures under white light to 100× magnification under oil immersion. Pictures were taken at 60× magnification. Examined areas in the brain were identified with images from the Allen Mouse Brain Atlas (2004) [99]. These anatomical reference atlases illustrate the adult mouse brain in coronal planes of section. They are the spatial framework for cell projection maps and in vitro cell characterization.

### 4.7. Identification of APP and GFAP Expression in the Brain Homogenate

Brains were grated in a homogeniser in immunoprecipitation buffer (RIPA) without SDS and deoxycholic acid at a ratio of 100 μL buffer per 10 mg tissue. Homogenates were kept for 1.5 h at 4 °C and then centrifuged at 3000 rpm for 15 min at 4 °C. Aliquots of the homogenates were frozen and stored at −80 °C until use. The total protein level was determined in the brain homogenate with Pierce BCA Protein Assay Kits (Thermo Fisher Scientific Inc., Waltham, MA, USA). Proteins were separated by SDS-PAGE and analysed by Western blotting for APP (amyloid-β precursor protein) and GFAP, which is a marker for gliosis. Protein separation was performed on a 4% thickening gel and a 10% separation gel. About 10 µg of protein was loaded onto the gel. The appropriate amount of the sample was mixed with the sample buffer and made up with RPIA to a volume of 20 µL. The samples were incubated for 5 min at 99 °C and centrifuged for 5 min at 14,000 rpm in an Eppendorf centrifuge, before loading onto the gels. Protein separation was carried out for 20 min at 80 V and then for 80 min at 100 V (constant). Protein bands were transferred on a PVD sheet (0.45 µm). The membrane was activated for 30 s in methanol and soaked in a transfer buffer. The transfer was carried out at 13 V for 1.5 h on ice. After transfer, membranes were washed briefly in TBST and blocked with 5% BSA for 1 h at 37 °C and then washed 3 times for 5 min in TBST. Primary antibodies: anti-Amyloid Precursor Protein (APP), specific for 27 amino acids (aa 99–126) of 99 amino acid-long transmembrane C-terminal fragment described as C99, the first amyloid β precursor (ab 32136, Abcam, Cambridge, UK) diluted at 1:1000; anti-Glial Fibrillary Acidic Protein (GFAP) (ab7260, Abcam, Cambridge, UK) diluted at 1:2000 and anti-Tubulin-α (B-5-1-2, Santa Cruz Biotechnology, Dallas, TX, USA) diluted at 1:500. These were suspended in 5% BSA in TBST. Membranes were incubated at 4 °C overnight and washed in TBST. The secondary antibodies used were HRP-conjugated goat anti-mouse IgG (sc-2005) and goat anti-rabbit IgG (sc-2004), Santa Cruz Biotechnology, at a dilution of 1: 5000 in 3% milk in TBST. The membrane was incubated for 2 h at room temperature and washed in TBST. The protein marker Pierce Prestained MW (26612), Thermo Fisher Scientific, with the range of 20–120 kDa was used.

Membranes were developed using a Thermo Scientific™ SuperSignal™ West Pico PLUS Chemiluminescent Substrate (No. 34580) chemiluminescent kit and read on a BioRadChemiDoc chemiluminescent instrument. The membranes were rinsed for 10 min in distilled water, then 10 min in 0.2 M NaOH solution and again in distilled water for 2 min. The above procedure was then repeated starting with the blocking. The densitometric analysis was performed using Image J Version 1.x (NIH). The protein level was determined on the basis of the light intensity of the bands. The samples were normalised compared with the reference protein (α-tubulin), for which the light intensity was set as 1.

### 4.8. Identification of Chitin-Like Structure in the Brain Section

Fluorescent Brightener 28 (Sigma-Aldrich) was used at a dilution of 1 mg/mL in distilled water to identify the chitin-like structure in neural tissue. Then, one drop of the dye and one drop of 10% potassium hydroxide were added and applied for 1 min. The tissue structure was contrasted by 1% trypan blue, rinsed in distillate water, mounted on microscope slides using 1% glycerol-gelatin and covered with cover-glasses. Slides were examined under a Nikon fluorescence microscope at 60× magnification under oil immersion and exposed to ultraviolet light for the DAPI channel.

### 4.9. Identification of Chitin-Like Polymers by Fourier Transform Infrared Spectroscopy Analysis

Samples of 200 µL of brain homogenate were diluted with 200 µL PBS with protein inhibitors. Samples were cooled to 4 °C, before being frozen at −20 °C until use. Attenuated Total Reflectance (ATR) analyses of the initial chitin were recorded on a Nicolet iS10 spectrophotometer (Thermo Scientific, Waltham, MA, USA) equipped with an attenuated total reflectance mode with a diamond as the crystal. The spectra were processed using Omnic 9.3.30 (Thermo Scientific, Waltham, MA, USA). For FTIR analysis, the initial chitin and N-acetyl-glucosamine and brain samples were placed on the spectrophotometric windows made of CaBr_2_ (Sigma-Aldrich, Poznań, Poland) and allowed to evaporate at room temperature (25 °C) for 24 h. FTIR spectra were collected in the wavelength range between 4000 cm^−1^ and 500 cm^−1^, at a resolution of 2 cm^−1^ using 100-times scanning.

All measurements were made on a CaF_2_ plate, onto which a vortexed sample drop was applied. After drying, the measurement was performed at room temperature, collecting spectra at several different locations in the sample. The results were analysed in relation to the two standards used, namely, N-acetyl-glucosamine by Sigma Aldrich (Poznań, Poland) and chitin by PRIMEX (Siglufjordur, Iceland). Finally, when compared with spectra from the literature [100,101,102,103], there is a great similarity to these spectra.

### 4.10. Determination of Survivin by ELISA

Survivin concentration was measured in the brain homogenate solution with the commercially available kit, ab202401-Survivin (Birc5) Mouse SimpleStep ELISA Kit (Abcam, Symbios, Gdańsk County, Poland), following the manufacturer’s instructions. The results were read at a wavelength of 450 nm.

### 4.11. Forced Swim Test

The day before the autopsy, a forced swim test was performed to identify unnatural movement patterns typical of depressed states, resulting from pathology in neural tissue caused by LPS and/or *H. polygyrus* infection. Mice were placed in an escapable transparent tank that was filled with water, and their escape-related mobility behaviour was measured.

The main measurement during the test was the number of seconds of immobility. In addition, no unusual behaviour was observed. The test took place in a testing room, and the test was carried out using glass cylindrical tanks (height 30 cm × diameter 20 cm). The tanks were filled with tap water set at room temperature (21–22 °C) to the determined level, which is marked on the tank walls. After preparing the tanks, the animals were brought into the testing room. The camera was placed on the top of the tank in order to obtain the highest possible visibility of the mice’s movements. Video recording was started before placing each animal into the water tank. Each mouse was tested individually. The animals were held by the tail and slowly placed in the water. Once a mouse was in the tank, we started the countdown on the stopwatch. Each mouse spent 6 min in the tank and was then pulled by the tail and put back into the cage. The test considered two parts: (1) 2 min of free-floating and (2) 4 min of behaviour measurement. Mice can readily float in water; however, most mice are very active at the beginning of the test. Therefore, the measurement should start omitting the first minutes, and usually, the behaviour is measured on the basis of the last 4 min of the test [104].

### 4.12. Statistical Analyses

The results collected from at least 5 mice of each group or 4 of APPSWE mice and statistical evaluation are presented. The significance of differences between groups was determined by Student *t*-test and analysis of variance (ANOVA) using MINITAB Software Minitab 21.1.0 (Minitab Inc., Philadelphia, PA, USA). All values were expressed as mean ± SE. A *p*-value of <0.05 was considered to be statistically significant. The specific *p*-value is indicated in the figures.

## 5. Conclusions

This paper highlights the potential link between intestinal nematode infection and neuroinflammatory and neurodegenerative processes induced by LPS, a potent inflammatory agent associated with TLR-4 activation.

The mouse model allowed us to identify neuroinflammatory and neurodegenerative markers in mice with impaired behaviour under stress conditions. Several results are especially relevant for further research. We have established that long-term nematode infection affects neuroinflammatory processes and causes neurodegeneration. Based on the immunoregulatory activity of many parasites, our results once again highlighted the highly complex interaction of parasite molecules with the host’s immune system, which can also modify host behaviour. Depending on the host’s pre-existing or underlying immune activity, parasitic infection may result in a different picture, such as changes in blood neutrophil response, the level of anti-apoptotic survivin, regulatory macrophage cell population and accumulation of pathogenic protein in the brain, or the response to stress conditions. Our results suggest that despite tissue destruction, neurons may be protected from neurodegeneration, which may be expressed in better antidepressant conditions, reaction to stress and moving skills. This is important for the evolutionary adaptation of the parasite, which can stay with a healthy host for a longer time. The better behavioural performance of mice infected only with nematodes than mice treated with LPS or genetically predisposed to neuropathological symptoms raises the question of molecules produced by adult worms that are involved in the protection of nerve cells and their function. The question also concerns the role of macrophages educated during parasitic invasion. Parasites can be the source of many factors that can be used as templates for the production of drugs to improve the health and mental condition of people with AD.

Preliminary studies on transgenic mice showed that the strain may be used to reflect parasite–host interaction in the AD model. Additional studies are needed to explain the brain–blood barrier permeability during neuroinflammatory disorders in this mouse model.

In the difficulties of successive treatment of people with chronic inflammatory diseases and due to the necessity of introducing effective anti-inflammatory drugs in transplant medicine, innovative drugs have to be introduced that can be natural products that do not cause side effects. Our study may be relevant to the medical treatment of patients with chronic inflammation. We hypothesised that immunoregulatory mechanisms operating during intestinal nematode infection may inhibit neurodegenerative changes provoked by LPS injection. We find that the outcome of *H. polygyrus*-dependent immunoregulation is different in healthy, control mice than in mice with existing LPS-induced inflammation. Infection with *H. polygyrus* protected control mice from some neurodegenerative changes in the brain and reduced depression symptoms, typical for ageing mice. In mice with LPS-induced inflammation and infected with the parasite, the neurodegenerative changes are accompanied by poorer scores of depression, and these were also present in genetically predisposed APPSWE mice. The results obtained from our experimental model for studying the impact of parasite-related immunoregulation on neuroprotection or inhibition of neurodegeneration do not confirm our hypothesis. Changes in macrophage populations suggest additional studies to explain brain–blood barrier permeability during neuroinflammatory disorders in this mouse model.

There may be some possible limitations in this study. Our studies reflect several phenomena not presented in previous studies. We provide several interesting results that may be relevant for brain inflammation and depression symptoms. Such knowledge comes from evaluating several markers typical for neurodegeneration in ageing hosts infected with an intestinal nematode.

The lengthy study (till 8 months age of mice) of blood cell smear showed that neutrophil percentages remain high during the infection. Monocytes, a potential source of tissue macrophages, dropped in all treated mice. These observations indicate the mobilization of bone marrow cells at a time when adult worms reside in the intestine and interact with the intestinal microbiota and may impact brain neuropathology through the gut–brain axis [105]. These phenomena might be of parasitological and microbiological interest.

In the cerebrospinal fluid of 12-month-old mice, we identified an increase in the population of M1 and M2 macrophages; M1 macrophages dominated in the LPS + HP group with worse behavioural patterns, but M2 in the HP group had the best FST performance. This suggests that the M2 macrophage population may be involved in the protection of neural tissue. This is preliminary information encouraging research on the mechanism of neuroprotection.

In the LPS + HP group, domination of M1, the lowest survivin level, positive chitin-like polysaccharide and increasing expression of short-chain proteins cleaved from long APP in the hippocampus and GFAP in both the hippocampus and the cortex characterised the pro-neurodegenerative milieu with a behavioural pattern of depression in mice. For the first time, we show that infection of mice with intestinal nematode may enhance systemic inflammation induced by LPS resulting in poorer neurodegeneration in the brain, resembling that of AD in humans. Thus, we propose a new animal model for studying the causes of neurodegeneration resulting from chronic diseases.

*Heligmosomoides polygyrus* predisposed mice to deposit chitin-like structures in the brain tissue. This novel result may be interesting for exploring the physiological mechanisms of the host–parasite metabolic interactions.

We also found several potential weaknesses in our study that, when resolved, may improve new research and experiment schedules.

The subject is new, and no specific data are available from the previous literature regarding brain neurodegeneration caused by infection with intestinal nematodes. The applied methodology is burdened with weaknesses, which in the future may be replaced by more innovative measurements.

Mainly, qualitative results are presented, and our research may be subject to sample measurement bias and selection bias. The ageing animals get neuropathology distinctly, which was shown by Western blot analysis. Figure 6C shows the number of mice with 100 kDa APP expression, but such analysis was not available for cleavage chains (24–28 kDa) because only a few positive results were identified in mice. In future studies, the number of samples needs to be greater to get more materials for better expression of results. The study can be repeated in other strains of mice to find out whether the changes occur because of the effectivity of immunomodulatory mechanisms triggered at the onset of infection or under the influence of factors excreted by adult forms of the parasite. C57BL/6 mice are susceptible to *H. polygyrus* infection, and in this strain, the innate inflammatory reaction is not sufficient to kill larvae in the intestinal wall. Adult worms may control immune responses by immunomodulation for a long (8 months) time, and the parasite activity might be positive for the downregulation of systemic inflammation.

We identified M1 and M2 macrophage populations in the brain fluid. Using other receptor markers could possibly distinguish the population of macrophages crossing the blood–brain barrier and local microglia cells [106]. These studies need separate experiments and would be profitable for the resolution of systemic immunoregulation mechanisms, affecting the brain.

Transgenic mice were added to the experiment as a positive control. Using transgenic mice in the parasitological experiment caused some problems due to the mortality rate of 20% among male mice and their aggressiveness. Building a control group of transgenic mice and a group injected with LPS would allow us to monitor how infection with *H. polygyrus* impacts the immunological response and genetically related neurodegenerative changes. Such studies need a new set of experiments and would be very interesting. It would be possible to identify if the intensity and time point of neurodegeneration are affected by the nematode infection in a stable genetic environment for AD.

Other methodologies, e.g., identifying separate cells and their physiological condition in the brain, may better reflect changes in a different part of the brain. We do not conduct quantitative analyses of astrocytes or neurons, because the slides were not cut from the same brain area. The methodology for histopathological evaluation needs to be improved and innovative tools developed for visualization of brain pathology.

Identification of chitin-like polysaccharides also needs quantitative evaluation and additional techniques to supplement FTIR.

The molecular explanation of distinct reactions in mice measured by the forced swim test should be explored in future studies, also using other parasitic infections.

## Figures and Tables

**Figure 1 ijms-24-13994-f001:**
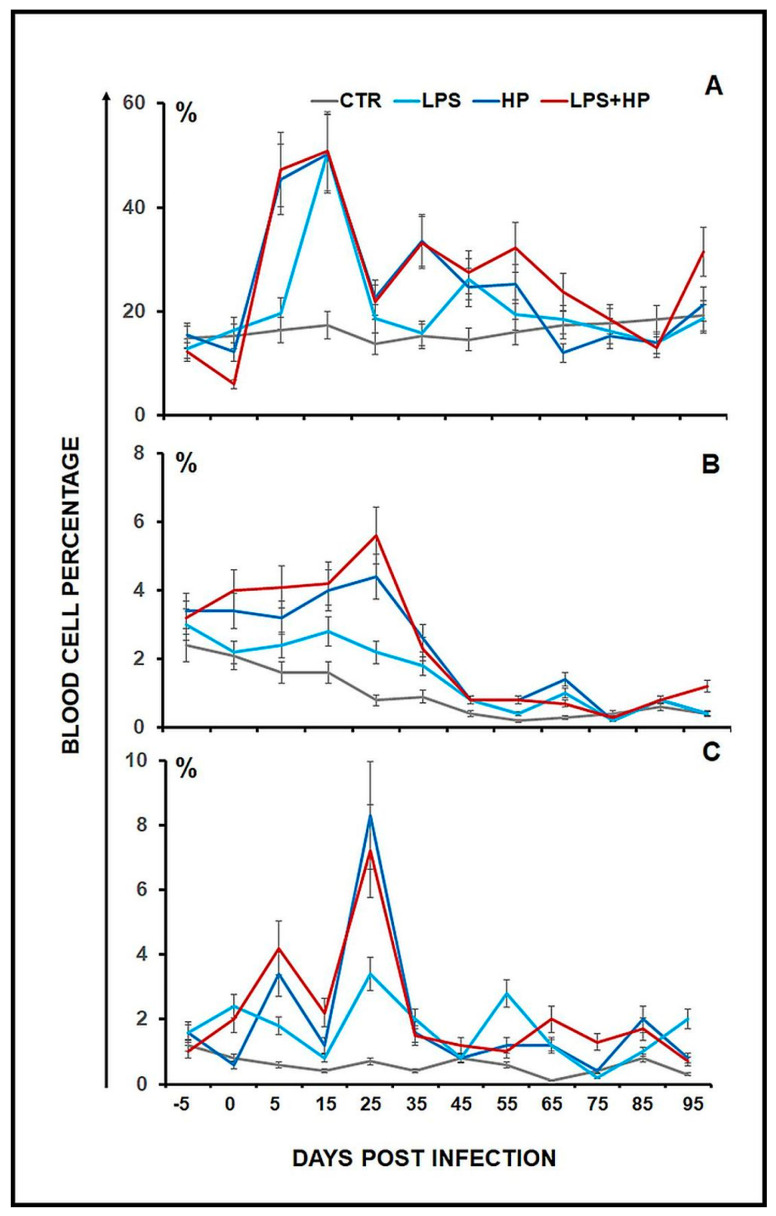
Changes in the percentage of neutrophils (**A**), monocytes (**B**) and eosinophils (**C**) in the blood smears of C57BL/6 mice, untreated, control (CTR), after injection with LPS (LPS), infection with *Heligmosomoides polygyrus* (HP) or injection with LPS and infection with *H. polygyrus* (LPS + HP). A statistically significant difference in the cell response: (**A**), ANOVA: between group, *p* < 0.002, the interaction between group and day, *p* < 0.0001; (**B**), ANOVA: between day, *p* < 0.000001; (**C**), ANOVA: between day, *p* < 0.00004, the interaction between group and day, *p* < 0.00002.

**Figure 2 ijms-24-13994-f002:**
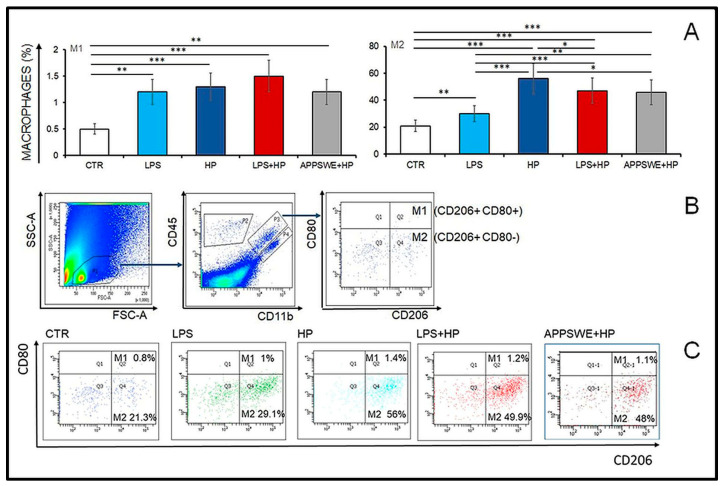
Changes in the cell population in the cerebrospinal fluid of C57BL/6 mice: untreated control (CTR), injected with LPS (LPS), infected with *Heligmosomoides polygyrus* (HP) or injected with LPS and infected with *H. polygyrus* (LPS + HP), and in APPSWE mice infected with *H. polygyrus* (APPSWE + HP). (**A**) Mean percentage of M1 and M2 macrophages. (**B**) Gating strategy for M1 and M2 analysis: all cells in the sample (P1), leukocytes negative for CD11b (P2), macrophages (P3) and microglia (P4) are distinguished based on the difference in CD11b and CD45 expression. Two populations of macrophages were identified in P3: M1 CD206+ CD80+ (Q2) and M2 CD206+ CD80− (Q4). (**C**) Representative dot plots. Statistical significance of differences between groups evaluated by Student *t*-test: * *p* < 0.05; ** *p* < 0.005; *** *p* < 0.00001. M1 ANOVA: between group, *p* < 0.0012; M2 ANOVA: between group, *p* < 0.00001.

**Figure 3 ijms-24-13994-f003:**
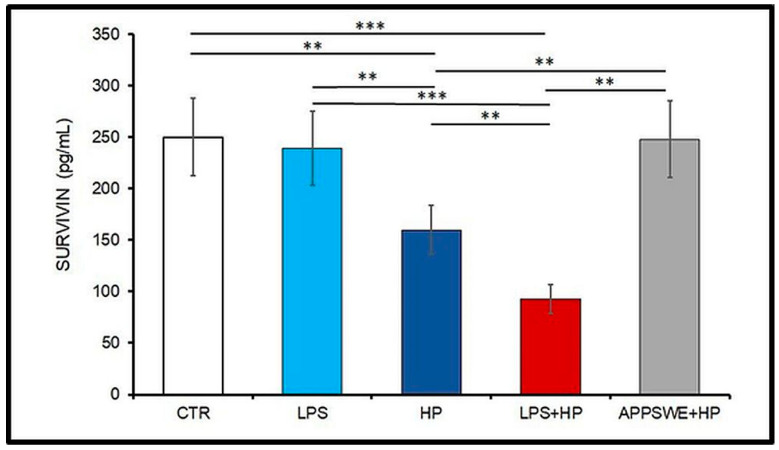
The concentration of survivin in the brain homogenate in C57BL/6 mice: untreated, control (CTR), injected with LPS (LPS), infected with *Heligmosomoides polygyrus* (HP) or injected with LPS and infected with *H. polygyrus* (LPS + HP)*;* APPSWE mice infected with *H. polygyrus* (APPSWE + HP). Samples were taken from 12-month-old mice infected with *H. polygyrus* at 4 months of age. Student *t*-test: ** *p* < 0.01; *** *p* < 0.001; ANOVA: between group, *p* < 0.000004.

**Figure 4 ijms-24-13994-f004:**
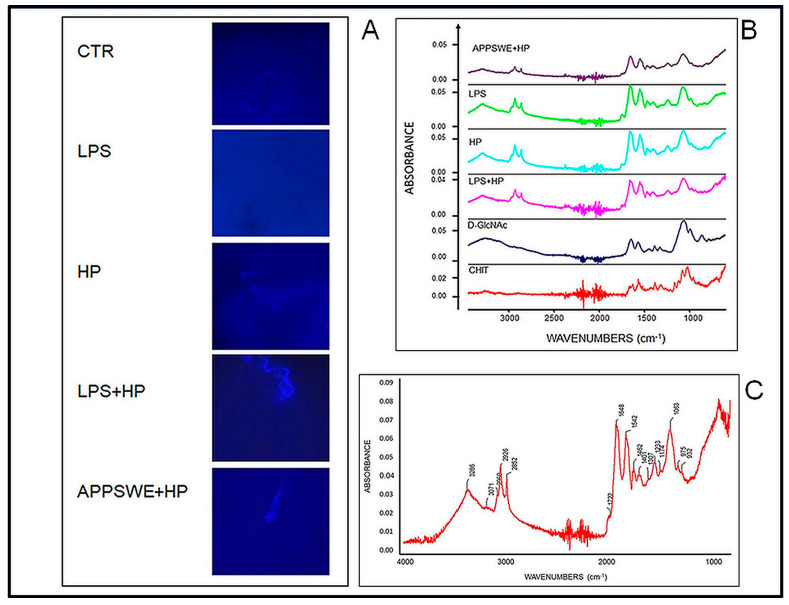
Chitin-like polysaccharides in the brain of C57BL/6 mice: untreated, control (CTR), injected with LPS (LPS); infected with *Heligmosomoides polygyrus* (HP); injected with LPS and infected with *H. polygyrus* (LPS + HP); APPSWE mice infected with *H. polygyrus* (APPSWE + HP). (**A**) Brain slides with the positive reaction for extracellular chitin-like structure labelled by Calcofluor (DAPI channel at 60× magnification, with immersion). Mice non-infected with the nematode were negative. (**B**) The comparative pattern of FTIR spectra for chitin-like polysaccharides in brain homogenate samples. (**C**) Chitin-like polysaccharides with spectra typical for chitin. Samples were taken from 12-month-old mice infected with *H. polygyrus* at 4 months of age.

**Figure 5 ijms-24-13994-f005:**
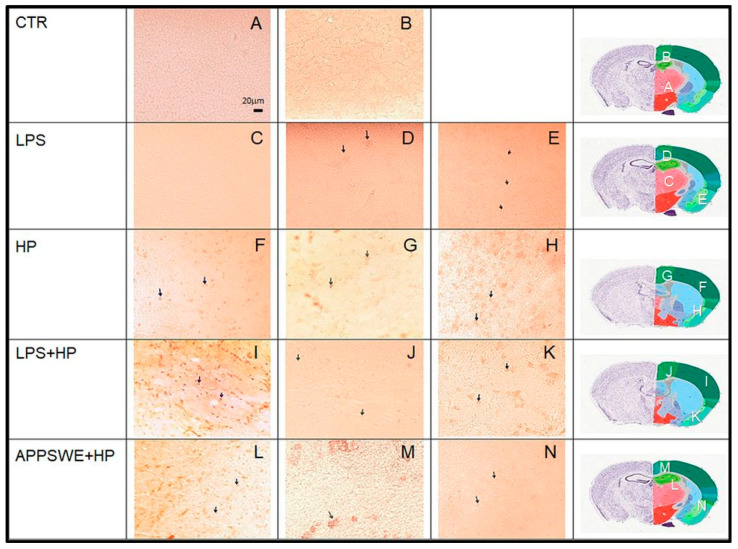
APP and GFAP co-detection in the brain of C56BL/6 mice: untreated, control mice (CTR); mice injected with LPS (LPS); mice infected with *Heligmosomoides polygyrus* (HP); mice injected with LPS and infected with *H. polygyrus* (LPS + HP) and APPSWE mice infected with *H. polygyrus* (APPSWE + HP). Atlas position of the tissue section in the brain (last column): thalamus (**A**,**C**), hippocampus (**B**,**L**), motor cortex (**D**,**G**,**J**,**M**), piriform cortex (**E**,**H**,**K**,**N**) and sensory cortex (**F,I**). Bright field, magnification 60×. The arrows in the LPS and HP groups indicate APP-positive neural cells. The arrows in the LPS + HP and APPSWE + HP groups indicate GFAP-positive astrocytes.

**Figure 6 ijms-24-13994-f006:**
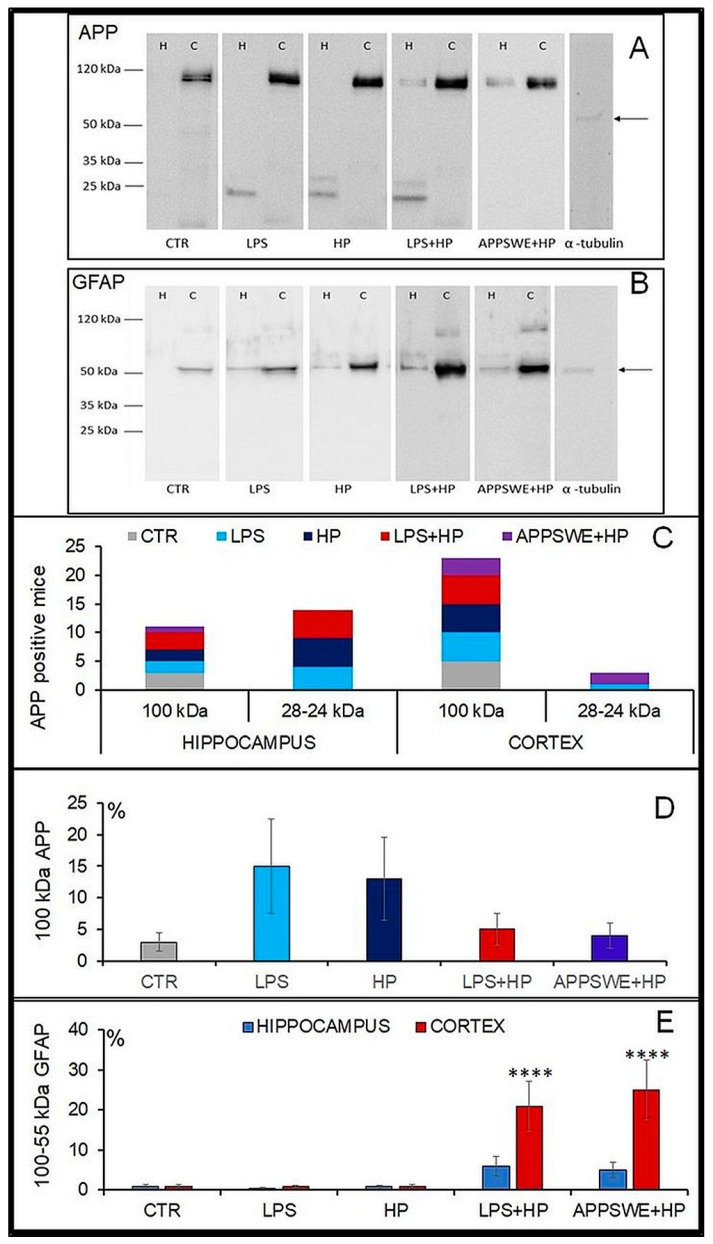
Western blotting for APP (**A**) and GFAP (**B**) in the brain of C57BL/6 mice: control mice (CTR), mice injected with LPS (LPS), mice infected with *Heligmosomoides polygyrus* (HP); mice injected with LPS and infected with *H. polygyrus* (LPS + HP) and APPSWE mice infected with *H. polygyrus.* The arrow indicates the α-tubulin position at 50 kDa. The rabbit anti-myeloid precursor protein (polyclonal antibodies AB5300 targeted against 27 amino acids (aa 99–126)) was used to detect the first amyloid β precursor; the typical set of bands corresponding to APP (95–100 kDa) in hippocampal (HIP) and cortex (COR) homogenates of 12-month-old mice. Additional smaller size bands in the hippocampus are also specific for APP, as they are not present in control (CTR) mice. They may be cytoplasmatic derivatives of APP with molecular weights between 24 and 28 kDa, indicating intensified protein cleavage. (**C**) Number of mice expressing long (100 kDa) and short (24–28 kDa) APP. The band protein was identified by Western blot and chemiluminescence. Not all mice in the group were positive in the cortex or hypothalamus for expression of short (24–28 kDa) APP chains. (**D**) Chemiluminescence intensity (%) of 100 kDa APP bands relative to 1 α-tubulin, a reference protein in brain tissue. The changes were not statistically significant. (**E**) Chemiluminescence intensity (%) of 55 kDa GFAP bands relative to 1 α-tubulin in brain tissue. The GFAP scores were statistically significant in the LPS + HP and APPSWE + HP groups in comparison to CTR, LPS and HP groups, in the cerebral cortex only. Student *t*-test, **** *p* < 0.0001; LPS + HP vs. CTR, LPS, HP; APPSWE + HP vs. CTR, LPS, HP.

**Figure 7 ijms-24-13994-f007:**
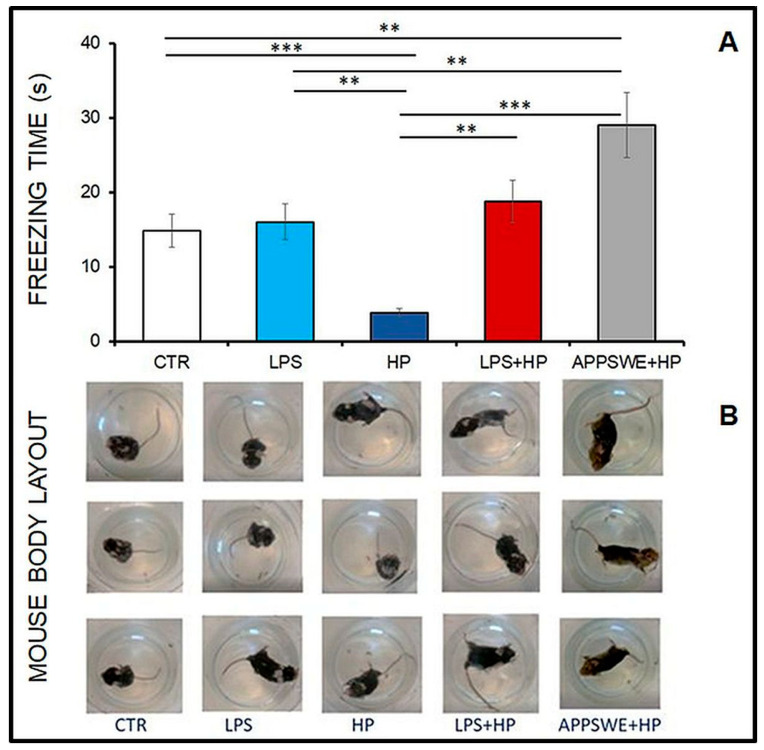
Behavioural changes in the forced swim test of C57BL/6 mice: untreated, control (CTR); injected with LPS (LPS), infected with *H. polygyrus* (HP); injected with LPS and infected with *H. polygyrus* and in APPSWE mice infected with *H. polygyrus* (APPSWE + HP). (**A**) Average number of seconds (± standard error) of freezing. Student *t*-test: ** *p* < 0.01; *** *p* < 0.001; ANOVA: between group, *p* < 0.0000003. (**B**) Behavioural pattern of mice in forced swim test: In FST, mice placed in a container of water cannot escape. First, the animals tried to escape (CTR-active movement) or were passive and did not swim, but floated in the water and tried to keep their nose above the water (LPS + HP, APPSWE + HP). The swimming positions of mice and mouse body layouts from the LPS and HP groups changed to active at different intervals.

## Data Availability

The original contributions presented in the study are included in the article. Further inquiries can be directed to the corresponding author.

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
