# Peer review of "Modulation of LPS-Induced Neurodegeneration by Intestinal Helminth Infection in Ageing Mice"

_ijms, 2023, doi:10.3390/ijms241813994_

Round 1
Reviewer 1 Report
In Abstract, the methods was long and the results were not specifically described. Abstract should be rewritten. In addition, because the intensity of APP, GFAP and chitin-like polysaccharide deposition was assessed but the neuronal survival or neuronal marker was not evaluated, "the degree of neurodegeneration assessed" was not appropriate.
Introduction might be too long.
In Results, line 192, how old of mice did the authors say "elderly mice"? In method section, the authors wrote that 20-25 g mice were used. The weeks or months old also should be added.
Figure 1, control group was absent. Moreover, what did the authors mean in the explanation of significant difference "between group and day"?
Figure 1-7, why was APPSWE (without HP infection) group absent? APPSWE should be added in groups to assess and compare the effects of HP infection.
In Results, line 239, the citation should be added for "The activity of astrocytes determines the survivin concentration in the brain".
Figure 4, the scale bar was lost in (A). And, why did not the authors perform the quantitative analysis?
Figure 5, the names of group were not shown in the figure. In addition, the figure was too small to understand cell morphology in A-N and the region in the right brain images.
In Results, line 329-330, "the amount of cleavage proteins increased both...."; the quantitative data was not present in Figure 6, therefore, the amount could not be judged. The quantitative data should be added.
Also in Figure 6, the lanes were separated between groups. It makes it impossible to quantify and compare the intensity. In Original Images for Blots/Gels, the explanations were absent and it was not understood the membranes were APP or GFAP, and each lane was which samples.
In Figure 7B, what did the three images of behavioral pattern of mice in each group show?
There were some spelling errors in the present manuscript.
Reviewer 2 Report
Jermakow and colleagues in the present research article entitled ‘The impact of Heligmosomoides polygyrus infection on the development of pathology in LPS induced neurodegeneration in C57Bl/6 mice’ investigate the impact of H. polygyrus infection on neuroinflammation and Alzheimer's disease (AD)-related markers in mice. The study focuses on understanding how the infection interacts with age-related neurodegenerative processes and inflammation, particularly concerning macrophage responses. The experimental design involves male C57BL/6 mice and a genetically modified strain (APPSWE) with Alzheimer's disease-like characteristics. The study reveals enhanced and prolonged inflammation in elderly mice subjected to LPS injection and H. polygyrus infection. M2 macrophages dominate the response, accompanied by reduced survivin levels in the brain. APP and GFAP expression is elevated, particularly in the hippocampus. M1 macrophages increase while M2 macrophages decrease in response to the infection, coinciding with lower forced swim test scores. Accumulation of pathologic precursors is observed in certain experimental groups, including APPSWE mice. Chitin-like polysaccharides accumulate in brain tissue due to H. polygyrus infection. The findings suggest that long-lasting H. polygyrus infection exacerbates neurodegeneration resulting from LPS-induced inflammation in the brains of elderly mice, mirroring aspects of AD pathology. Greater M2 macrophage populations appear to attenuate neuron pathogenesis in the brain cortex. The infection's immunoregulatory function requires further exploration, particularly in relation to brain-blood barrier permeability during neuroinflammatory disorders. The study underscores the complex interplay between neuroinflammation, parasitic infection, and AD-like markers. H. polygyrus infection exacerbates inflammation-induced neurodegeneration, potentially via modulation of macrophage responses. These findings offer insights into the intricate relationship between parasitic infections, immune responses, and neurodegenerative processes.
Overall, the research highlights several key findings and implications related to the interaction between Heligmosomoides polygyrus (a parasitic nematode) infection, lipopolysaccharide (LPS) injection-induced inflammation, and brain pathology in mice. In general, I think the idea of this article is really interesting and the authors’ fascinating observations on this timely topic may be of interest to the readers of International Journal of Molecular Sciences. However, some comments, as well as some crucial evidence that should be included to support the author’s argumentation, needed to be addressed to improve the quality of the manuscript, its adequacy, and its readability prior to the publication in the present form.
Please consider the following comments:
• I recommend revising the title. In its current form, I find it to be relatively long and could be made more concise without losing its essential elements. I would suggest rephrasing or abbreviating certain parts of the title to make it more succinct. A potential revised title could be: " Modulation of LPS-Induced Neurodegeneration by Heligmosomoides Infection in Mice." [1-3].
• A graphical abstract that will visually summarize the main findings of the manuscript is highly recommended.
• Abstract: According to the Journal’s guidelines, this section should be presented as a short summary of about 200 words maximum that objectively represents the article. It should let readers get the gist or essence of the manuscript quickly, prepare the readers to follow the detailed information, analyses, and arguments in the full paper and, most of all, it should help readers remember key points from your paper. Please, consider rewrite this paragraph following these instructions [4].
• Keywords: Please list ten keywords chosen from Medical Subject Headings (MeSH) and use as many as possible in the title and in the first two sentences of the abstract. I would suggest adding “Neuroinflammation” and “microglia” as keywords.
• Introduction: The authors need to reorganize this section with several paragraphs made up of about 1000 words, introducing information on the main constructs of this study, which should be understood by a reader in any discipline, and making it persuasive enough to put forward the main purpose of the current research the author has conducted and the specific purpose the author has intended by this protocol. I would like to encourage the authors to present the introduction starting with the general background, proceeding to the specific background on the connection between inflammation and Alzheimer's disease. Those main structures should be organized in a logical and cohesive manner [5].
• In this regard, I believe that the Introduction section would benefit from additional information to enhance its clarity and contextualization. To strengthen this section, I suggest highlighting the understanding of neural substrates that are known to underlie neurodegeneration, with a focus on Alzheimer's disease (AD). Neurodegeneration, as observed in conditions such as AD, involves the progressive and irreversible loss of neural structure and function. This paper delves into the intricate link between neurodegeneration and AD, which is characterized by the aggregation of misfolded proteins, primarily β-amyloid and tau, leading to the formation of plaques and neurofibrillary tangles, respectively. These aberrations disrupt neural communication and give rise to cognitive deficits [6-7]. Furthermore, highlighting the current understanding of the cellular and molecular mechanisms driving neurodegeneration can provide essential context. Oxidative stress, mitochondrial dysfunction, and excitotoxicity are known factors that contribute to the demise of neural cells. Additionally, the interplay of genetics and environmental factors, as well as the inflammatory responses mounted by the brain in the face of injury or infection, further contribute to this intricate process. Integrating these aspects into the introduction would provide a clear expanse of the complex phenomena underlying neurodegeneration and AD. This overview would not only align the readership with the subsequent sections but would also underscore the significance of the study in contributing to this field of knowledge [8-10].
• Results: In my opinion, the section on the changes in the immune cell population in the blood is informative. However, consider adding more context to the significance of these changes. Are the observed alterations in immune cell populations consistent with previous studies or hypotheses?
• Material and Methods: I believe that this section would benefit from a clearer structure and better organization of the flow of information. For example, I believe that the section should include information about the rationale behind using male C57BL/6 mice and mice with Alzheimer's disease (B6;SJL-Tg(APPSWE)2576Kha) for the experiments. Why were these strains chosen? What relevance do they have to the research questions? Also, I believe that here authors should explain the reasoning behind monitoring peripheral inflammation by examining the percentage of different leukocyte populations. How does this relate to the research objectives?
• In my opinion, the ‘Conclusions’ paragraph would benefit from some thoughtful as well as in-depth considerations by the authors, because as it stands, it lists down all the main findings of the research, without really stressing the theoretical significance of the study. Authors should make an effort, trying to explain the theoretical implication as well as the translational application of their research.
• In according to the previous comment, I would ask the authors to include a proper and defined ‘Limitations and future directions’ section before the end of the manuscript, in which authors can describe in detail and report all the technical issues brought to the surface,
• References: Authors should consider revising the bibliography, as there are several incorrect citations. Indeed, according to the Journal’s guidelines, they should provide the abbreviated journal name in italics, the year of publication in bold, the volume number in italics for all the references.
I hope that, after these careful revisions, the manuscript can meet the Journal’s high standards for publication. I am available for a new round of revision of this article.
I declare no conflict of interest regarding this manuscript.
Best regards,
Reviewer
References:
1. https://plos.org/resource/how-to-write-a-great-title/
2. https://www.nature.com/nature-index/news-blog/how-to-write-a-good-research-science-academic-paper-title
3. https://www.indeed.com/career-advice/career-development/catchy-title
4. https://www.mdpi.com/journal/ijms/instructions
5. https://dept.writing.wisc.edu/wac/writing-an-introduction-for-a-scientific-paper/
6. DOI: 10.17219/acem/165944
7. https://doi.org/10.3390/ijms24065926
8. DOI: 10.3390/biomedicines11030945
9. https://doi.org/10.3389/fnmol.2023.1217090
10. https://doi.org/10.3390/biomedicines11051248
Minor editing of English language required.
Round 2
Reviewer 1 Report
The reviewer could understand the authors' response and revisions.
There were some typographical and grammatical errors.
Author Response
Dera Editor,
Thank you for your comments and help in preparing a better edition of our manuscript. A molecular biologist (native English) corrected the manuscript.
All changes are written in red in the attached file.

Reviewer 2 Report
Dear Authors,
I am pleased to acknowledge that you have indeed addressed all of my concerns and queries in a clear and precise manner. Your responses have provided valuable insights into the modifications made to the manuscript in light of my comments. It is evident that you have taken great care to ensure that the revised manuscript aligns more closely with the scientific rigor expected for publication in IJMS.
Upon reviewing the updated version, I find that the inclusion of the additional studies has indeed enriched the understanding of neural substrates associated with neurodegeneration, particularly in the context of Alzheimer's disease (AD). The provided studies contribute significantly to the comprehensiveness of the section. However, in order to provide a more holistic view of the complex phenomena underlying neurodegeneration and AD, I believe there's still an opportunity to expand upon certain factors. Specifically, the discussion of oxidative stress, mitochondrial dysfunction, and excitotoxicity as known contributors to the demise of neural cells would offer a deeper insight into the mechanisms at play (DOI: 10.3390/biomedicines11030945; https://doi.org/10.3389/fnmol.2023.1217090; DOI: 10.3390/cells11162607). By incorporating these aspects, the Introduction section would offer a comprehensive overview of the multifaceted processes driving neurodegeneration. This, in turn, would provide readers with a clearer understanding of the intricate nature of AD.
I want to reiterate my appreciation for your responsiveness and willingness to consider these suggestions. I believe that this minor revision will significantly enhance the quality and impact of the Introduction section.
Thank you once again for your dedication to improving the manuscript. I look forward to seeing the continued progress.
Best regards,
Reviewer
Author Response
Thank you for your advice and suggestions. The "introduction” has been supplemented with publications that indicate an essential aspect of neurodegeneration, i.e. the multifactorial nature of the process. Thank you for choosing the publications.
The paragraph in the introduction in lines: 46-60
“Neurodegenerative diseases are associated with systemic and local intra- and extracellular changes; activated astrocytes and microglia produce nitric oxide (NO), reactive oxygen species and pro-inflammatory cytokines. IL-1β, IL-6 and TNF-α levels are elevated peripherally and in the brain [3,4]. Glial cell-derived mediators increase the level of cytokines, while immune cells of peripheral origin have been identified during neuroinflammation [5,6]. Oxidative stress contributes to neurodegeneration by causing damage to axons and neurons. In addition to oxidative stress and immune-mediated inflammatory responses, glutamate excitotoxicity and mitochondrial dysfunction also play a role in the pathogenesis and progression of neurodegenerative diseases. During demyelination, the concentrations of 5-hydroxytryptamine, tryptophan (TRP) and kynurenine (KYN) metabolites are changed [7] and contribute to the development of pathological conditions including neurological and psychiatric disorders [8]. The increased levels of the KYN metabolite in the brain have been associated with altered fear states resulting from trauma, stress, and anxiety [9]. “
We are very grateful for helping us improve our manuscript.
